# Biomimetic Biomolecules in Next Generation Xeno-Hybrid Bone Graft Material Show Enhanced In Vitro Bone Cells Response

**DOI:** 10.3390/jcm8122159

**Published:** 2019-12-06

**Authors:** Giuseppe Perale, Marta Monjo, Joana M. Ramis, Øystein Øvrebø, Felice Betge, Petter Lyngstadaas, Håvard J. Haugen

**Affiliations:** 1Industrie Biomediche Insubri SA, Via Cantonale 67, 6805 Mezzovico-Vira, Switzerland; oystein676@gmail.com (Ø.Ø.); felice.betge@ibi-sa.com (F.B.); 2Ludwig Boltzmann Institute for Experimental and Clinical Traumatology, Donaueschingenstrasse 13, 1200 Vienna, Austria; 3Cell Therapy and Tissue Engineering Group, Research Institute on Health Sciences (IUNICS), University of the Balearic Islands. Ctra. Valldemossa km 7.5, 07122 Palma de Mallorca, Spain; marta.monjo@uib.es (M.M.); joana.ramis@uib.es (J.M.R.); 4Balearic Islands Health Research Institute (IdISBa), 07010 Palma de Mallorca, Spain; 5Corticalis AS, Oslo Sciencepark, Gaustadallen 21, 0349 Oslo, Norway; lyngstadaas@corticalis.com (P.L.); haugen@corticalis.com (H.J.H.)

**Keywords:** bone graft, scaffold, xeno-hybride, biomolecule, intrinsic disorder protein

## Abstract

Bone defects resulting from trauma, disease, surgery or congenital malformations are a significant health problem worldwide. Consequently, bone is the second most transplanted tissue just after blood. Although bone grafts (BGs) have been used for decades to improve bone repairs, none of the currently available BGs possesses all the desirable characteristics. One way to overcome such limitations is to introduce the feature of controlled release of active bone-promoting biomolecules: however, the administration of, e.g., recombinant Bone morphogenetic proteins (BMPs) have been used in concentrations overshooting physiologically occurring concentrations and has thus raised concerns as documented side effects were recorded. Secondly, most such biomolecules are very sensitive to organic solvents and this hinders their use. Here, we present a novel xeno-hybrid bone graft, SmartBonePep^®^, with a new type of biomolecule (i.e., intrinsically disordered proteins, IDPs) that is both resistant to processing with organic solvent and both triggers bone cells proliferation and differentiation. SmartBonePep^®^ is an advanced and improved modification of SmartBone^®^, which is a bone substitute produced by combining naturally-derived mineral bone structures with resorbable polymers and collagen fragments. Not only have we demonstrated that Intrinsically Disordered Proteins (IDPs) can be successfully and safely loaded onto a SmartBonePep^®^, withstanding the hefty manufacturing processes, but also made them bioavailable in a tuneable manner and proved that these biomolecules are a robust and resilient biomolecule family, being a better candidate with respect to other biomolecules for effectively producing the next generation bone grafts. Most other biomolecules which enhances bone formation, e.g., BMP, would not have tolerated the organic solvent used to produce SmartBonePep^®^.

## 1. Introduction

A frequently used method to treat bone defects is the use of bone grafts to promote tissue regeneration. This as it provides an osteoconductive, osteoinductive and/or osteogenic environment that promotes bone repair and healing [1]. Currently autografting, the transplantation of tissue from one site of the patient’s body to another, represents the gold standard [2,3,4]. However, this method has drawbacks such as limited supply and
the risk of donor site morbidity, particularly in oncological applications, and hardly applies in paediatric cases [5,6]. Moreover, it has been reported that 31% of patients still experience persistent pain at the donor site 2 years after operation [7]. This has motivated the search for alternative bone grafts and different sources, where particularly synthetic grafts made of biopolymers, bioceramics and their composites have received a lot
of research focus and industrial interest lately [8,9,10,11,12,13]: there’s a common census within the research community on the need of next generation bone grafts with higher bioactivity degree [6,14]. However, the synthetic grafts traditionally tend to lag behind on the biological performance compared to autografts, allografts and xenografts [15,16]. This said, a former study comparing sinus floor augmentation with inorganic bovine bone (Bio-Oss^®^, Geistlich AG, Wolhusen, Switzerland) and biphasic calcium phosphate (Straumann^®^ BoneCeramic, Institute Straumann AG, Basel, Switzerland) showed no statistical difference between the bone formation for the two groups [15], proving that commercial research on synthetic bone grafts are catching up. Looking at the academic research environment, many of the cited studies use additive manufacturing systems to produce their grafts. Although 3D printing has a promising potential, particularly when it comes to customized implants, the implementation is still limited by high production cost and long production time [17] and yet no medical devices manufactured with this technology have reach such a maturity to gain robust clinical data to allow being certified and hence being safely put into the market. The current commercial market for bone grafts primarily consists of devices based on the refinement of bovine mineral matrix or cadaver derived tissues: an exemplificative set of examples for treatment of orthopaedic defects are e.g., SmartBone^®^ (I.B.I. SA, Mezzovico Vira, Switzerland—xeno-hybrid graft), Bio-Oss^®^ (Geistlich, Lucerne, Switzerland —bovine xenograft) and MaxGraft^®^ (Biotiss, Zossen, Germany—processed human allograft). Noteworthy, one of the main advantages with the xeno-hybrid bone grafts is their ability to resorb and remodel, in addition to have high intrinsic mechanical strength.

Even though there are many commercially available bone-substitutes, none is specifically addressing either the paediatric or the oncologic indications. Besides referring to critical patients, these types of applications share a common clinical feature, i.e., the need to support bone remodelling in a critical environment, either for the growth pressure in kids, or for the presence of chemotherapeutic agents that usually impair or reduce bone physiologic metabolism. Literature robustly suggest that fractures represent 20%–30% of all bone-related diagnosis for children [16]. This means that there is an increasing treatment demand with increased population size. The paediatric market is unique as the skeletal immaturity of the patients is linked to considerable further bone growth [18], inducing very high performance requirements to such medical devices. During puberty, the human body experiences a very high increase in physical dimensions, however the increase in bone mass seems to lag behind [16]. Similarly, paediatric oncologic cases impacting bones are worldwide increasing, both for primary and secondary tumours [18]: primary bone tumours, 0.2% of all cancers, register yearly ca. 3500 new cases per 400M population, with a 3:2 male/female ratio, and record almost 45% deaths; secondary bone tumours incidence statistics are almost two orders of magnitude higher. One of most common approach to bone cancer is the combined approach of either chemotherapy or irradiation and surgical removal; this last creates the need of reconstructing a bone segment portion of a priori not precisely known dimensions. Literature suggests controversial results for traditional grafting materials [19].

The use of bioactive molecules has arisen as a promising alternative to assist in challenging bone regeneration. A number of growth factors and other biomolecules that regulate this complex physiology, including BMPs, transforming growth factor (TGF-β), insulin-like growth factor (IGF), fibroblast-like growth factor (FGF), platelet derived growth factor (PDGF) and vascular endothelial growth factor (VEGF), have been suggested to be used alone or in combination with a bone graft material for therapeutic use in bone regeneration [20,21,22]. Bone graft growth mechanism with stimulating biomolecule is very complex; however, it is advantageous to develop a device that regulates the bone regeneration in a manner that stimulates biomineralization at a controlled manner. Moreover, the long-term effects of bone morphogenic proteins (BMPs), the most abundantly used growth factor for bone tissue regeneration, are not clearly identified; this also prevents it from being FDA-approved for paediatric treatment [23,24] and they are discouraged in oncologic patients [25]. This leads to the demand for a new type of growth factor or signalling system, with improved properties and high biocompatibility.

Lately, it has been observed that intrinsically disordered proteins (IDPs) play a crucial role in biomineralization, particularly through signalling and regulation of the direction and extend of mineral crystal growth [26]. This has brought up the question of the importance of proline in mineralization. Enamel matrix derivative (EMD) is an extract of enamel matrix and contains amelogenins (AMEL) of various molecular weights [27,28]. AMEL are an IDPs and are involved in the formation of enamel and periodontal attachment formation during tooth development [29]. AMEL are hydrophobic proteins rich in proline aminoacid that regulate initiation and growth of hydroxyapatite during enamel formation [30,31]. AMELs self-assemble into supramolecular aggregates and precipitate to form an extra cellular matrix layer [32] that acts as an extracellular matrix scaffold for the growing crystallites [29]. Emdogain^®^ is a commercial product that has been approved for clinical use in the regeneration of periodontal attachment and bone, (Straumann AG, Basel, Switzerland), where the major constituents are AMEL dissolved in a hydrogel. A large number of clinical studies now confirms the clinical effect of Emdogain^®^ [33,34,35]. The peptide sequence P6 used in this study is identical to part of the AMEL sequence [36,37]. Rubert et al. [31] showed structural and biological activity for synthetic proline-rich peptides, which initiated similar osteoblastic differentiation compared to a commercial product Emdogain^®^ (Straumann AG, Basel, Switzerland). The major constituents of Emdogain^®^ are amelogenins [38,39,40].

These evidences drove us towards the belief that, by including finely optimised synthetic proline-rich peptides in a bone graft, we can produce a bone graft with superior biological activity with respect to the current commercially available solutions, hence matching the still unmet needs also in critical spaces such as paediatric reconstructions and oncology [6].

The present work was aimed to develop and assess the in vitro performances of a xeno-hybrid composite bone graft, enriched with proline-based peptides, also named SmartBonePep^®^ (SBP) and to verify that the added peptide had a positive outcome on cellular response. A murine osteoblastic cell line MC3T3-E1 (DSMZ, Braunschweig, Germany) was used for the in vitro experiments. In vitro studies with large samples numbers can be screened rapidly with the fast growing MC3T3-E1 cell line and this cell line provides reproducible results and is robust and well applicable for screening purposes [41]. The cell line is much used in initial studies of biomaterials intended for use in bone application [42,43,44,45,46,47], nevertheless, for studies concerning detailed involved bone formation mechanisms, cell lines are not sufficient. In comparison, primary cells have a better indication for the expected effects in vivo, but are also less stable and employ different pathways as well as differences in expression of proteins and mRNA levels [48].

## 2. Experimental Section

### 2.1. SmartBonePep Development

SmartBonePep^®^ (SBP) is a further development of the commercially available xeno-hybrid bone graft SmartBone^®^ (SBN): SBN consist of a bovine bone-derived mineral matrix which is improved by reinforcement with the co-polymer coating poly(L-lactide-co-ε-caprolactone) (PLCL) and the addition of RGD-exposing collagen fragments from animal-derived gelatine (manufactured by I.B.I. SA, Mezzovico-Vira, Switzerland). SBN is described in detail in previous works [49,50,51]. In SBP, the NuPep peptide is embedded in the polymer coating of SBN: SBP indeed includes the same components and same composition as the SmartBone^®^ but with the IDPs (commercially traded as NuPep, by Corticalis AS, Oslo, Norway) physically entrapped within the polymeric coating during manufacturing process which was not modified nor altered in any step [34]. Two different NuPep sequences, namely P2 and P6, were here tested, targeting a punctual 1 μg/cc concentration of NuPep within SBP graft. Their structures are summarized in Table 1. Experiments have been performed using SBN as control versus SBP with target concentration of either P2 (SBP2) or P6 (SBP6), and combined P2 and P6 at target concentration (SBP2-P6) and P6 at 1/10 of target concentration (SBP6–1/10). The target concentration is set by previous studies [36,37,52,53] with NuPep and is equivalent to molar concentration of EMD in Emdogain^®^. This set-up was designed to eventually highlight the presence of cross-effects between P2 and P6.

### 2.2. NuPep Loading and Release

A model molecule, Fluorescein Isothiocyanate Isomer I (FITC), has been used to assess correct loading and release profiles before using NuPep. Data are provided in supporting information. The same method was used for making the release profile of NuPep from SBP. To identify the presence of NuPep in the aqueous solution via infrared spectroscopy, FITC molecules were either attached to the C- or N-terminal of the peptide before loading. Labelled peptides, indeed, show a similar steric hindrance to bare peptides and hence diffusion coefficient and release kinetics can be reasonably expected as comparable.

### 2.3. Graft Production, Samples Release and Basic Characterization

For practical sake of testing reasons, and thanks to manufacturing availability to obtain different shapes and sizes, SBP samples were manufactured in disc shape, 3 mm in height and 16 mm in diameter, for in vitro testing and in 10 × 10 × 10 mm^3^ cubes for release and mechanical testing. The larger samples for cell culture was made to ensure better cell spreading. The cubes for mechanical testing was made as these cubes are similar in size to those BS used in the clinic. All manufacturing batches were released following same SBN release procedures, under c-GMP and ISO13485–2016 compliance (by Industrie Biomedicine Insubri SA, Mezzovico-Vira, Switzerland). Graft microstructural characterization was performed using analytical methods in correspondence with standard literature [54]: extended pressure electronic microscopy (*a.k.a.* environmental scanning electron microscope (SEM), alias E/SEM) and with energy dispersion spectroscopy (EDS) with a JEOL 6010-LA SEM (Jeol, Tokyo, Japan) using 15 kV, at ×100 magnitude. Mechanical testing was performed according to standard literature [50].

All groups (*n =* 3), were scanned using micro-CT (Bruker microCT1172, Kontich, Belgium) with the scanning parameters set at 104 μA, 95 kV, image rotation of 0.400 degrees and a resolution of 7.9 μm with an aluminium copper filter. Three dimensional models were reconstructed in the software CTvox (Bruker, Kontich, Belgium) and exemplificative snapshots were obtained therefrom. The structure parameters were calculated using the software CT Analysis (Bruker microCT, Kontich, Belgium).

### 2.4. In Vitro Testing, Cell Cultures

Discs of 16 mm in diameter and 3 mm in height of SBP and SBN were used for the studies. Discs had 4 different codes related to the different synthetic peptides, plus control: SBP2, SBP2-P6, SBP6, SBP6–1/10 and control SBN.

Mouse osteoblastic cell line, MC3T3-E1, was obtained from the German Collection of Microorganisms and Cell Cultures (DSMZ, Braunschweig, Germany). MC3T3-E1 cells were routinely cultured at 37 °C in a humidified atmosphere of 5% CO_2_ and maintained in α-MEM supplemented with 10% foetal calf serum (FCS) and antibiotics (50 IU penicillin/mL and 50 µg streptomycin/mL). Cells were sub-cultured 1:4 before reaching confluence using PBS and trypsin/EDTA. All experiments were performed in the same passage of the MC3T3-E1 cells.

To test the different BGS surfaces and formulations, these were placed in a 12-well plate and 2 × 10^5^ cells were seeded on each disc. In order to guarantee a homogenous cell distribution inside BGS, an agitated seeding method was used [55]. Briefly, after adding 1 ml of cell suspension to the BGS, plates were agitated on an orbital shaker (Unitron, Infors HT, Basel, Switzerland) for 6 h at 180 rpm at 37 °C and in humidity conditions. Then, cells were maintained in static conditions at 37 °C in a humidified atmosphere of 5% CO_2_ for up to 14 days. The same number of cells were cultured in parallel in plastic in all the experiments.

For the experiments, MC3T3-E1 cells were maintained for 72 h (toxicity, scanning electron microscopy) and 14 days (gene expression analyses, alkaline phosphatase (ALP) activity, SEM and confocal microscopy) on the BGS in α-MEM supplemented with 10% FCS and antibiotics.

### 2.5. Determination of Cell Viability: LDH Activity

Lactate dehydrogenase (LDH) activity in the culture media was used as an index of cell death. LDH activity was determined spectrophotometrically after 30 min incubation at 25 °C of 100 µL of culture and 100 µL of the reaction mixture by measuring the oxidation of NADH at 490 nm in the presence of pyruvate, according to the manufacturer’s kit instructions (Roche Diagnostics, Mannheim, Germany). Results were presented relative to the LDH activity in the medium of cells seeded on TCP (low control, 0% of cell death) and on TCP adding triton X-100 1% (high control, 100% of death), using the Equation (1):Cytotoxicity (%) = (epivalve – low control)/ (high control – low control) × 100(1)

### 2.6. Immunofluorescence

Cells grown for 14 days on SBP different samples and on SBN controls were fixed for 15 min with 4% formaldehyde in PBS at room temperature. For nuclei visualization, cells were stained with DAPI (Sigma, St. Louis, MO, USA). Then, few drops of Fluoroshield™ (Sigma, St. Louis, MO, USA) were added and cover glasses were mounted on the samples. Two samples of each group were used to perform the experiment and two images of each sample were taken with the confocal microscope (Leica DMI 4000B equipped with Leica TCS SPE laser system).

### 2.7. Scanning Electron Microscopy

Cells grown for 72 h or 14 days on SBP different samples and on SBN controls were fixed with glutaraldehyde 4% in PBS for 2 hrs. The fixative solution was removed, and the cells were washed with distilled water two times. At 30 min intervals, the cells were dehydrated by the addition of 50%, 70%, 90% and 100% ethanol solutions. Finally, the ethanol was removed, and the cells were left at room temperature to evaporate the remaining ethanol prior to analysis. Two samples of each group were used to perform the experiment and two images of each sample were taken with the scanning electron microscope (Hitachi S-3400N, Hitachi High-Technologies Europe GmbH, Krefeld, Germany).

### 2.8. Real-time PCR Analysis

Total RNA was isolated with Tripure (Roche Diagnostics, Nonnenwald, Germany) and RNA was quantified using a spectrophotometer set at 260 nm. Real-time PCR was performed for two reference genes and several target genes (Table 1).

The same amount of total RNA from each sample (900 ng) was reverse transcribed to cDNA at 37 °C for 60 min in a final volume of 20 µL, using High Capacity RNA to cDNA kit (Applied Biosystems). Each cDNA was diluted 1:10 and these dilutions were used to carry on the quantitative PCR.

Real-time PCR was performed in the Lightcycler 480^®^ (Roche Diagnostics, Nonnenwald, Germany) using SYBR green detection. For each reaction, 7 μL of Lightcycler-FastStart DNA MasterPLUS SYBR Green I, 0.5 μM of sense and antisense specific primers and 3μL of the cDNA dilution in a final volume of 10 μL were added. The normal amplification program consisted of a preincubation step for denaturation of the template cDNA (95 °C), followed by 45 cycles consisting of a denaturation step (95 °C), an annealing step (60 °C, for all except for osterix, which was 68 °C and an extension step (72 °C). Real-time efficiencies were calculated from the given slopes in the LightCycler 480 software (Roche Diagnostics, Nonnenwald, Germany) using serial dilutions.

Relative quantification after PCR was calculated by dividing the concentration of the target gene in each sample by the mean of the concentration of the two reference genes (also known as housekeeping genes, Table 2) in the same sample using the advanced relative quantification method, provided by the LightCycler 480 analysis software version 1.5 (Roche Diagnostics, Nonnenwald, Germany).

### 2.9. Alkaline Phosphatase (ALP) Activity

Alkaline phosphatase (ALP) activity was determined from cells after 14 days of cell culture. Cells were washed twice in PBS, solubilized with 0.1% Triton X -100. Samples were then incubated with an assay mixture of p-Nitrophenyl Phosphate (pNPP). Cleavage of pNPP (Sigma, Saint Louis, Missouri, Mo, USA) in a soluble yellow end product which absorb at 405 nm was used to assess ALP activity. In parallel to samples, measurement, a standard curve with calf intestinal alkaline phosphatase (CIAP) (Promega, Madison, Wi, USA) was constructed; 1 µl from the stock CIAP was mixed with 5 mL of alkaline phosphatase buffer (1:5000 dilution), and subsequently diluted 1:5.

### 2.10. Statistics

All data are presented as mean values ± SEM. The Kolmogorov-Smirnov test was done to assume parametric or non-parametric distributions for the normality tests. Differences between groups were statistically compared by ANOVA and Bonferroni as a post hoc test. Results were considered statistically significant at *p*-values <0.05. SPSS^®^ program for Windows, version 17.0 (SPSS Inc, Chicago, IL, USA) was used.

## 3. Results

### 3.1. Analytical Investigation on Peptide Loading and Release

Release tests performed via UV-vis spectroscopy confirmed that both the chromophore and the labelled peptides (version P2 and P6, with FITC attached on the C or the N terminals) were properly loaded onto SBP, confirming the feasibility of loading bare NuPep in a similar manner. Experimental data on SBN loaded with the chromophore are reported in the supporting information, while release data of labelled-NuPep are presented in Figure 1. It may observe that both the two labelled peptide versions (-C and -N labelled terminals), show similar release profiles, confirming a pure Fickian release [56].

### 3.2. SBP Grafts Characterization

SBP samples were analysed according to standard procedure for SBN and resulted compliant for release. Indeed, no residual solvents used during manufacturing were detected by gas chromatography. Furthermore, E/SEM imaging and energy dispersive spectroscopy (EDS) spectra confirmed the correct presence of the polymeric coating. Exemplificative images of SBP surfaces are presented in Figure 2. When comparing these data with previous literature [50,51], E/SEM study confirmed an identical microstructure for SBN and SBP. The polymer phase is recognized by the carbon content (evidenced in the EDS spectra) compared to the bone phase. Likewise, the bone phase has calcium and phosphorous content. Bone largely consists of crystalline apatite [57], which has a molecular structure consisting both of calcium and phosphorous.

Figure 3 presents exemplificative images of internal surfaces of SBP, exposed by halving the samples [43,49], which all showed equivalent results on the internal surfaces as the external, indicating a good dispersion of the polymer coating. Overall, these evidences from external and internal surfaces confirm that the coating was successfully applied.

Mechanical performances are summarized in Table 3 and, considering the animal origin of the base matrix, they resulted statistically identical to those of SBN [50].

Given the peptides concentration in the coating is extremely low, all macroscopic properties of SBP are hence confirmed being identical to those of SBN.

Micro-CT results showed that there was no significant structural difference between SBN and SBP, including open porosity, pore diameter, structural separation, and surface/volume ratio (Table 4). 

### 3.3. In Vitro Results

The biocompatibility of the different groups of SBP was first evaluated quantitatively utilizing LDH activity, versus SBN and positive (C+) and negative (C-) controls (Figure 4). The LDH assay detects the amount of LDH that leaks out through the plasma membrane of damaged cells, as a marker of cytotoxicity. As it can be observed in Figure 4, after 72 h of cell culture, all SBP groups had significantly higher toxicity than low control after 3 days of incubation, and cytotoxicity values were over 30%, which is the maximum value accepted for cytotoxicity of medical devices according to current ISO-10993:5, although often considered debatable as this test is not always reliable.

Indeed, afterwards cells grew well onto and inside the scaffolds as observed by E/SEM imaging (Figure 5) and confocal images (Figure 6). Exemplificative pictures are presented in Figure 5, taken after 3 and 14 days in culture, respectively; considerable morphological changes of the seeded cells on the surface of the graft can be observed overtime. After three days, the cells are barely observable on the graft surface, however after 14 days there has been excellent cell growth for all groups, and particularly for SBP2 and SBP6, with widely spreading cells in different layers.

Cell ingrowth inside the scaffolding material was investigated using confocal microscopy of stained samples (Figure 6) after two days of culture, although few cells were observed on the SEM images, confocal microscopy confirmed their presence and penetration through the SBP 3D matrix. This was probably due to the attachment and growth of the cells onto the polymer of the SBP samples, which impairs their visualization under the SEM. After 14 days of culture, cells grew well onto the surface of all the SBP groups and inside the scaffolds. However, confocal images confirmed the presence of cells through at least 60 µm in depth showing that cells seeded on the SBP2 group penetrated more deeply (pink nuclei can be observed both at 2 and 14 days) compared to SBP6 (no pink nuclei can be observed). Noteworthy, it should be pointed that the method used here might not reflect the situation in the centre of the scaffold.

Regarding gene expression of cell adhesion and extracellular matrix markers analysed an up-regulation of the mRNA levels of collagen type 1 was observed in the SBP6 group, although no statistical differences were found. This trend agrees with the higher number of cells observed by E/SEM (Figure 5) and total RNA content (Figure 7), although this is a qualitative observation and not a direct measure of the number of cells and, indeed, it should be considered as such (Figure 8 and Figure 9).

Regarding the gene expression of selected osteoblast differentiation markers (Figure 8 and Figure 9), at day 14, the mRNA expression levels of all the evaluated markers were increased either in the SBP6 group (Bone sialoprotein, Bsp, an early differentiation marker) or in the SBP6–1/10 group (Bone morphogenetic protein 2, Bmp2; Osterix, Osx; Osteocalcin, Oc) although only statistical significance was reached for the osteocalcin gene, showing higher expression values for the SBP6–1/10 group compared to SBN, SBP2 and SBP6. Noteworthy, no significant differences among groups were found for ALP activity.

## 4. Discussion

Bone grafts have been predominately used to treat bone defects, delayed union or non-union fractures; however, the ideal bone graft substitute is still lacking [6,14]. One way to improve the biological response to bone graft material is to introduce various growth factors, such as bone morphogenetic proteins (BMPs), parathyroid hormone (PTH) and platelet rich plasma (PRP), into structural allografts and synthetic bone substitutes [58,59,60]. Although clinical applications of these factors have exhibited good bone formation, their further application was limited due to high cost and potential adverse side effects [61]. Inorganic ions such as magnesium, strontium and zinc are considered as alternative of osteogenic biological factors [59].

A class of biomolecules that has gain a lot of attention in the literature is represented by intrinsically disordered proteins (IDPs), which have a crucial role in biomineralization, particularly through signalling and regulation of direction and extend of bone mineral crystal growth [26]. This by affecting chemical and cellular events such as cell signalling, macromolecular self-assembly, protein removal and crystal nucleation and growth [62]. IDPs are recognized by that they are in whole or in part heterogeneously ensembled of flexible molecules, in an unorganized manner, causing its 3D structure to be undefined [62,63], hence disordered. This makes them highly flexible molecules and recent results suggests that IDPs with tuned disorder-order ratio can be used to program the mineralization process, yielding growth of aligned nanocrystals into hierarchical mineralized structures [64]. A protein can be characterized as mostly disordered and fully disordered if more than 50% and 90%, respectively, of its residues is locally disordered [26]. Some of these peptides has already been used commercially in medical devices: e.g., Straumann AG (Basel, Switzerland) produces and sells the commercially known Straumann^®^ Emdogain^®^, a hydrogel primarily based on the enamel matrix derived protein (EMD) [65] amelogenin (AMEL) [66,67,68]. EMD/AMEL mimics the biological process of natural tooth development and enables the regeneration of new periodontal tissue. In 1997, Hammarstrom et al. [69,70] found that EMD promotes the reformation of acellular cementum and alveolar bone, and ever since a mass of clinical studies have been confirming EMD clinical effect [71,72,73]. NuPep replicates sequences found in AMEL [74,75].

Although some IDPs have been used commercially, e.g., the Emdogain^®^ just seen, NuPep (NP) is to the authors’ best knowledge the first synthetic IDP that is incorporated into a bone graft material showing positive results on cell line. NP is an artificial peptide with several “disorganised” proline-rich regions. Both amelogenin and ameloblastin, two IDPs important for enamel mineralization, contains proline-rich regions [37]. This has brought up the question of the importance of the proline in mineralization. Rubert et al. [37] already showed that these peptide structures initiated osteoblastic differentiation compared similar to the biological motifs of EMD. Another important factor is that NP can withstand processing within organic solvents, unlike most other biomolecule used to enhance bone graft’s biological performance. Indeed, in the manufacturing of SBP, strong organic solvents are used, coming from SBN standard manufacturing process and needed to solve the aliphatic polyesters composing the thin polymeric film [33]. It has already been established that SBN had a microstructure similar to healthy dense spongy human bone, with an average of 63% porosity, and that that the rigged structure of the SBN ensures better osteointegration than just a bovine based xenograft [33,50,51]. By analysing the SBP coating, it has been confirmed that the microstructure is equivalent to that of SBN and there was a homogenous distribution of the peptide as one observed a uniformed spread during EDX analysis. This is a so-called statistically homogeneous coating structure, as it is meant to mimic natural tissues that do not show homogenous structures [76]. Ideally, growth factors should promote tissue growth and mineralization such that the regenerated tissue has similar structure to the native tissue environment [6]. This may be difficult with homogenous distribution of conventional growth factors such as BMP or EDM, however NP has been designed to have an automatic biological react on stimuli from its environment and would fold accordingly; this makes these special IDPs like an “automatic on-and-off function” and allows the bone graft carrying it to be active only when required.

Thus, overall, the growth mechanism the bone graft should mimic is very complex: there is necessity to develop a device that regulates the bone regeneration in a manner that stimulates biomineralization at a high rate, but only when necessary. Moreover, the long-term effects of bone morphogenic proteins (BMP), the most abundantly used growth factor for bone tissue regeneration, are not clearly identified, which prevents it from being FDA-approved for paediatric treatment [23,24] and discouraged in oncologic patients [25]. This leads to the demand for a new type of growth factor, or signalling system, with improved properties and high biocompatibility.

Here, we not only demonstrated that NuPep can be successfully loaded but also made them bioavailable in a tuned manner, proving they are a robust and resilient biomolecule family that can be effectively used also in industrially scaled-up manufacturing processes, successfully withstanding solvent processing and sterilization too.

Indeed, bone requires its different cells to be at different stages of proliferation, differentiation and maturation in a multi-layered organised structure to promote successful tissue growth [77]. Therefore, it is important to tune the peptide release rate such that there is a high early growth, but also reassure stage diversity for the cells. The target release for the peptide bioavailability was 1 μg per cc SBP per day for about one week. Considering that release is a Fickian-based phenomenon [37], this should give a cumulated release of about 12–14 μg over 14 days, with a release burst in the first 2 days. As seen in the results, P2 had its highest release rates the two first days and after 14 days was the release of P2 around 11.75–14.01 μg. This confirms that the release was as designed, hence according also to the safety-by-design paradigm, and most relevantly as biologically desired. Hence, with a clinical translational perspective, SBP has a peptide release rate, which is very adequate to sustain robust and fast bone regeneration in critical applications. One must, however, mention that these releases where done in purify water and that one could expect a different profile once placed in an in vivo environment, due to possible interactions with ions and proteins.

Considering the mRNA level associated with NP release, there is a strong indication that the peptide presents higher osteoblast differentiation and promotes osteogenesis. Moreover, the significantly higher mRNA than the amelogenin derived EMD illustrates the enormous potential of IDP when it comes to the clinical success of tissue regeneration. Particularly, poor osteointegration and tissue formation of most current available graft is driving the need for enhanced growth and improved yield of newly formed bone [77]. Therefore, the integration of NP in SBN, i.e., the SBP, is a good step in this regenerative direction. Indeed, all SBP groups tested in these experiments increased LDH activity released to cell culture media in MC3T3-E1 cells after 72 h of cell culture compared to cells cultured onto tissue culture plastic. Moreover, E/SEM and confocal images after 14 days of culture showed a good osteoconductivity, with a higher number of cells in the SBP6 group observed by E/SEM. These results are in agreement with the higher amount of total RNA that was found from SBP6 samples compared to all the other groups and a trend to increased collagen type 1 mRNA levels. The best gene expression profile for osteoblast differentiation was found for the SBP6–1/10 group, while no significant differences among groups were found for ALP activity. Taking together all the results, the SBP6 group induces a higher degree of proliferation of MC3T3-E1 cells while the SBP6–1/10 is the group that promotes a higher osteoblast differentiation.

## 5. Conclusions

Bone is the second most transplanted tissue, after blood, and bone grafts are predominately used to treat bone defects, delayed union or non-union fractures. In a framework where the ideal bone graft substitute is still lacking, major efforts are taken in the direction of composite solutions also with various growth factors, with the final aim to improve the biological response and hence support remodelling and, finally, new tissue formation.

Here, we investigated two candidate biomolecules from the promising family of intrinsically disordered peptides, namely the P2 and P6 sequences of the NuPep library. In order to let them express their best osteogenic potential, we embedded them into a bone scaffold, specifically a high performance xenohybrid bone substitute with a relevantly positive track record of clinical successes, SmartBone^®^. Not only we had demonstrated that NuPep can be successfully and safely loaded onto SmartBone^®^, withstanding heavy manufacturing processes that include the use of aggressive solvents and sterilization, but also made them bioavailable in a tuned manner, according to the desired controlled delivery pattern. We recorded enhanced bone cells response from in vitro investigations and, overall, we proved that IDPs are a robust and resilient biomolecule family, being a better candidate with respect to other biomolecules for effectively producing the next generation bone grafts. 

## Figures and Tables

**Figure 1 jcm-08-02159-f001:**
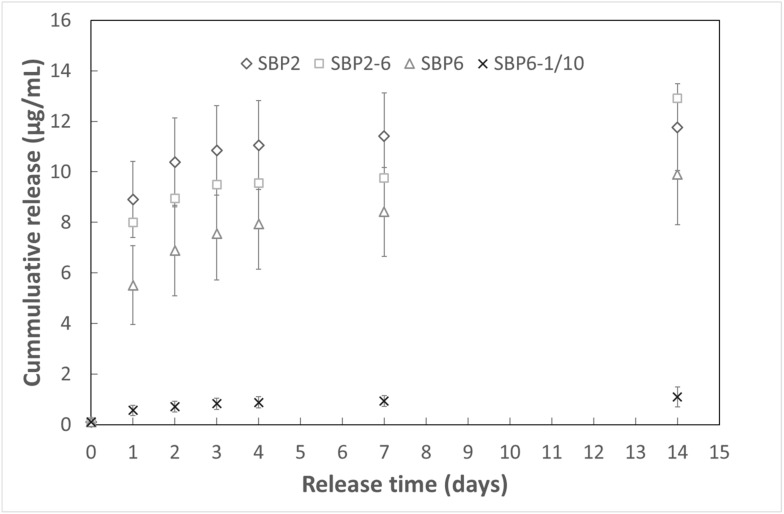
Released mass of NuPep in μg versus time in days for the different bone graft substitute in 10 mL purified water.

**Figure 2 jcm-08-02159-f002:**
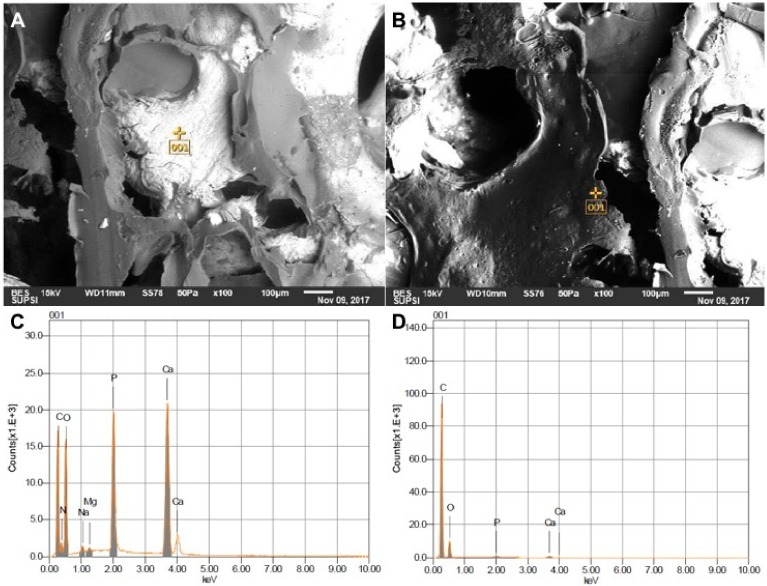
(**A**) and (**B**): E/SEM analysis of random point on the external surface of the SmartBonePep^®^ (SBP) grafts. (**A**) and (**C**): E/SEM and EDS of bony phase. (**B**) and (**D**): E/SEM and EDS of polymeric phase.

**Figure 3 jcm-08-02159-f003:**
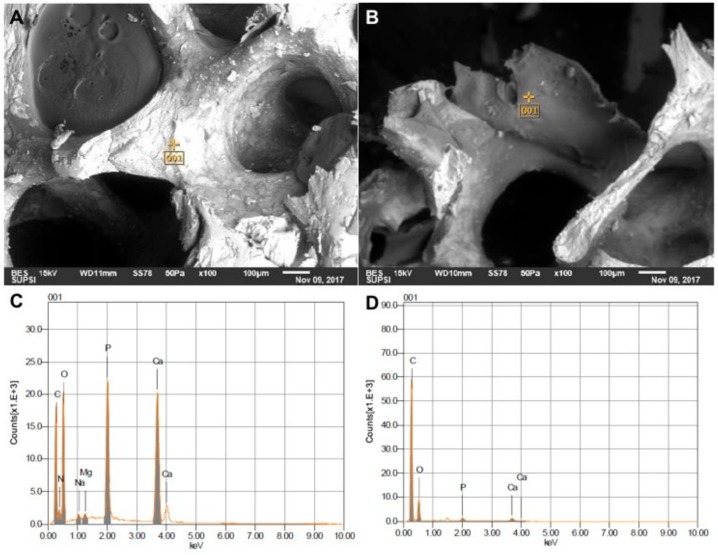
(**A**) and (**B**): E/SEM analysis of random point on the internal surface of the grafts. (**A**) and (**C**): E/SEM and EDS of bony phase. (**B**) and (**D**): E/SEM and EDS of polymeric phase.

**Figure 4 jcm-08-02159-f004:**
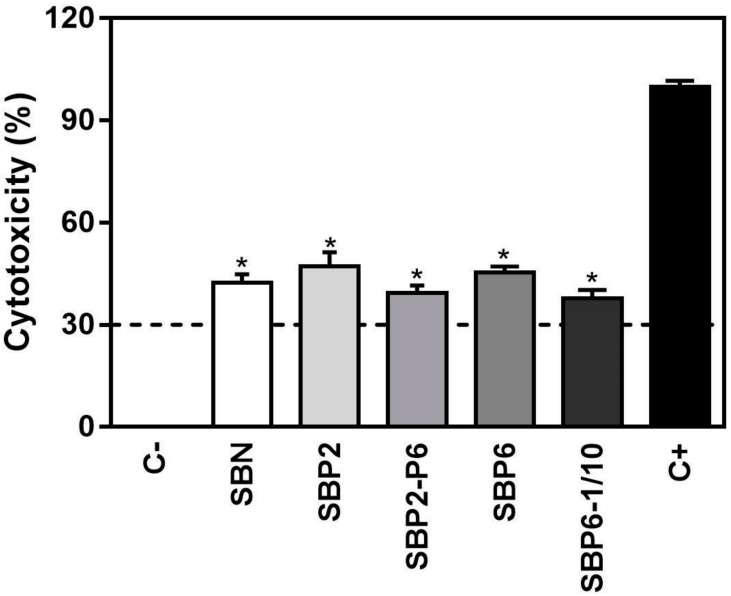
Lactate dehydrogenase (LDH) activity of SPB samples tested, versus SB and positive (C+) and negative (C-) controls. Lactate dehydrogenase (LDH) activity, is an indicator of cytotoxicity and was here measured in culture media after 72 h of incubation with the test samples and controls. Low control (C-, 0% toxicity) was obtained from culture media of MC3T3-E1 cells seeded on plastic (TCP). High control (C+, 100% toxicity) was obtained from culture media of MC3T3-E1 cells seeded on TCP and treated with 1% Triton X-100. Five groups of SBP were tested, versus SBN: SBP2, SBP2-P6, SBP6 and SBP6–1/10. Values represent the mean ± SEM. One-way ANOVA and DMS post hoc analyses were performed: * *p* < 0.05 vs. C-.

**Figure 5 jcm-08-02159-f005:**
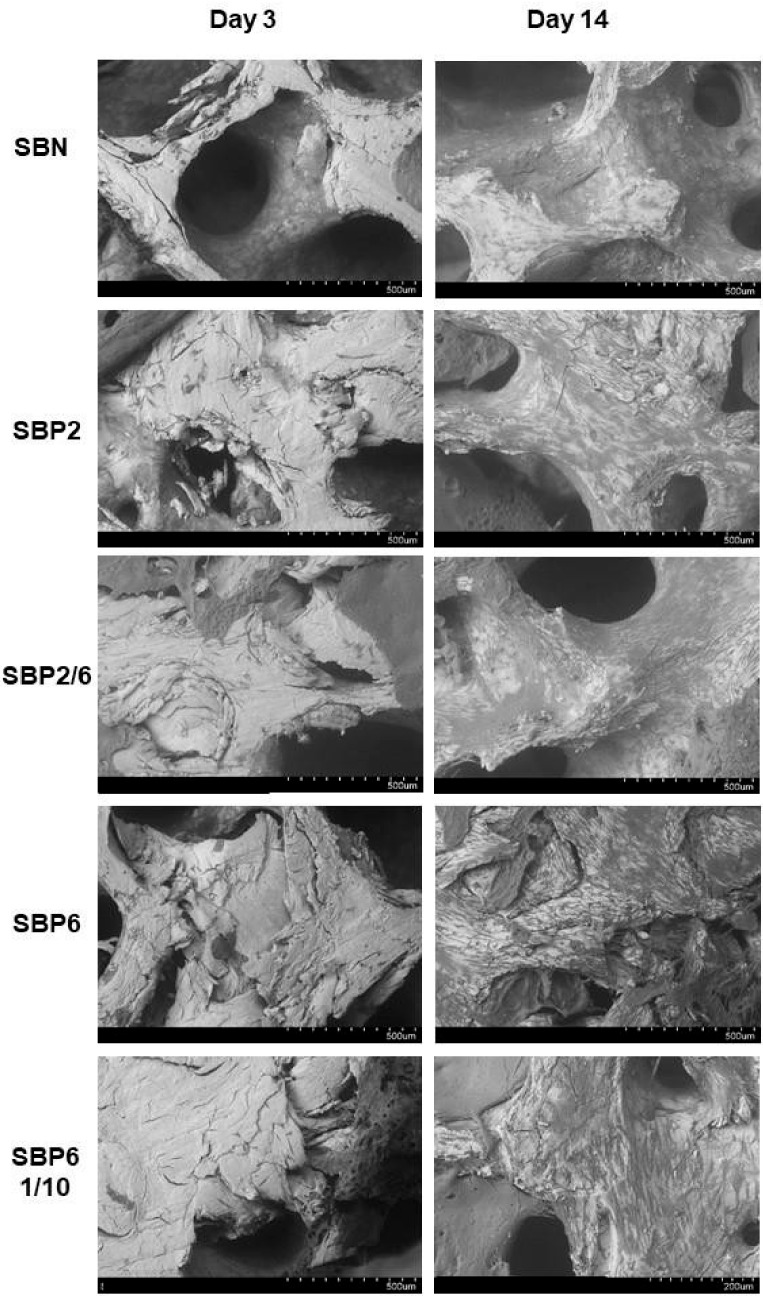
E/SEM visualization of MC3T3-E1 cells growing on the surface of SBP samples for 3 and 14 days. SBP samples were observed by SEM at 10 kV, 40 Pa using back scattered and secondary electrons. Exemplificative images presented are from representative superficial areas randomly taken.

**Figure 6 jcm-08-02159-f006:**
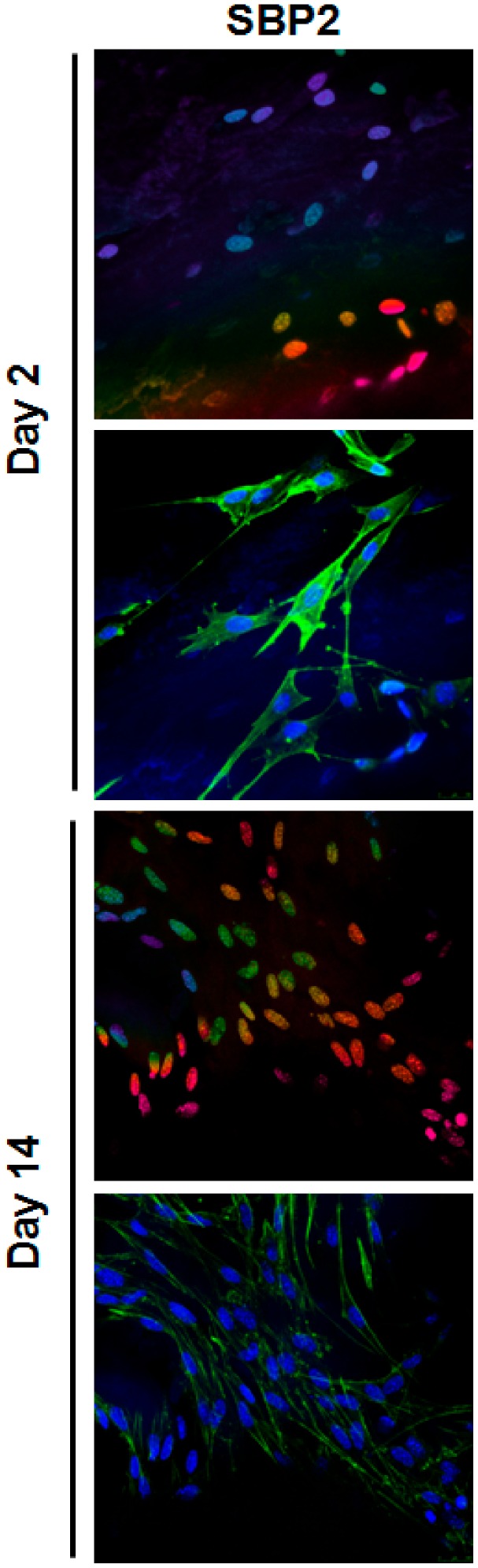
Confocal micrographs of MC3T3-E1 cells cultured for 2 and 14 days on SBP samples. Cells were stained with Phalloidin-FITC (stains actin filaments; green) and DAPI (stains nucleus; blue). The upper row of each day shows depth projection micrographs, images are 2D reconstructions of sections acquired repeatedly in sequential steps along the z-axis. The colour code on these rows corresponds to the z-axis depth of DAPI-stained nucleus, coded from blue at 0 µm and pink at 60 µm depth.

**Figure 7 jcm-08-02159-f007:**
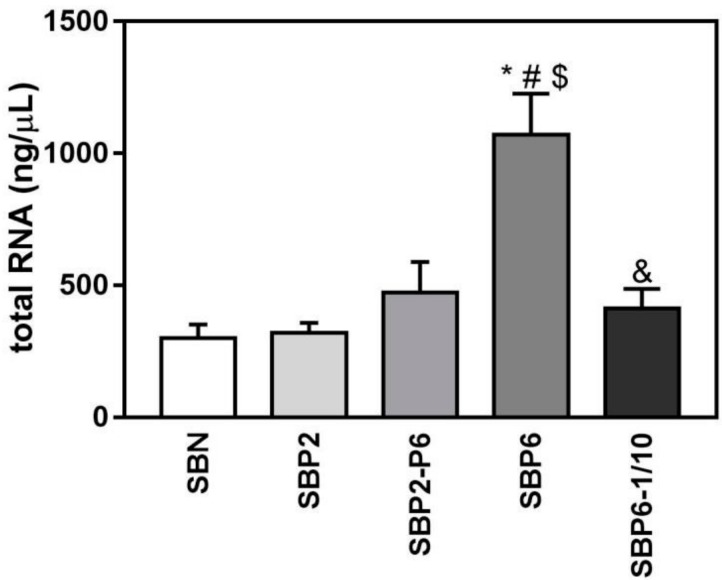
Total RNA extracted from the cells seeded on all SBP samples at 14 days of cell culture, versus SBN. Values represent mean ± SEM. One-way ANOVA and DMS post hoc analyses were performed: * *p* < 0.05 vs. SBN, # vs. SBP2, $ vs. SBP2-P6, and vs. SBP6.

**Figure 8 jcm-08-02159-f008:**
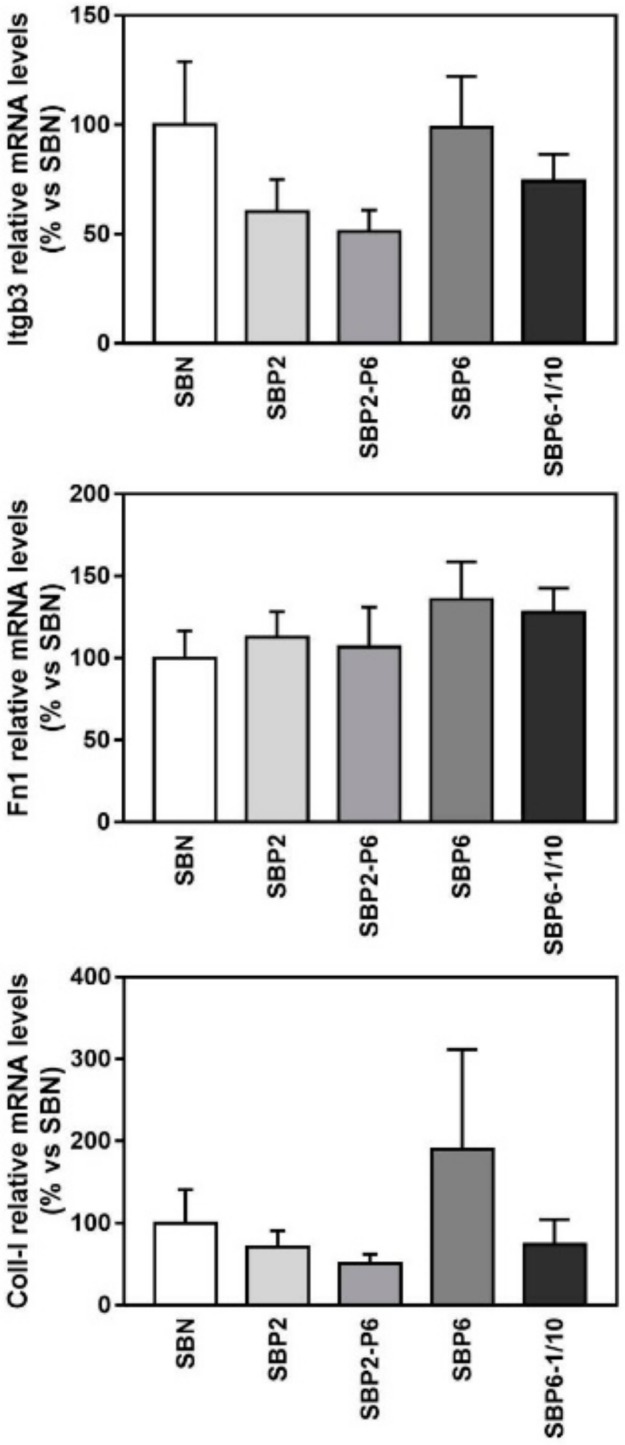
Relative mRNA levels of Integrin beta 3 (Itgb3), Fibronectin 1 (Fn1) and Collagen type 1 (Coll-I) in MC3T3-E1 cells cultured on seeded on all SBP samples at 14 days of cell culture, versus SBN. Data represent fold changes of target genes normalized with reference genes (rRNA 18 S and Gapdh), expressed as a percentage of cells seeded on SBN, which were set to 100%. Values represent mean ± SEM. One-way ANOVA and post hoc analyses were performed. No significant differences were found.

**Figure 9 jcm-08-02159-f009:**
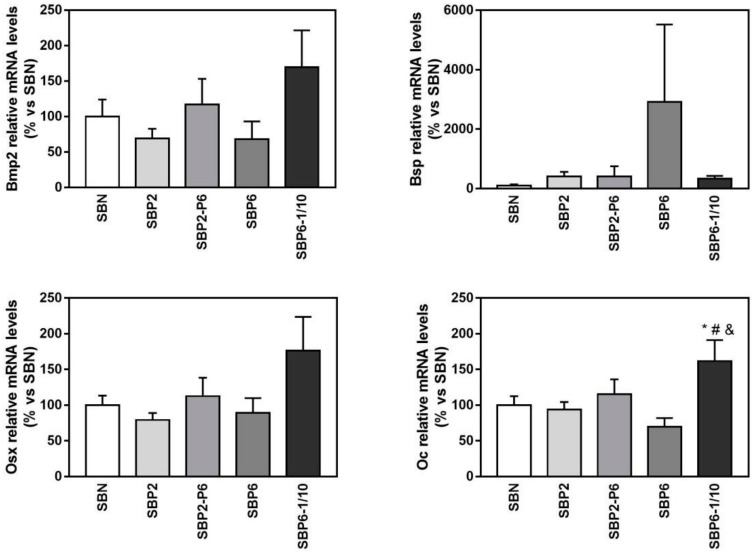
Relative mRNA levels of Bone morphogenetic protein 2 (Bmp2), Bone sialoprotein (Bsp), Osterix (Osx), and Osteocalcin (Oc) in MC3T3-E1 cells cultured on seeded on all SBP samples at 14 days of cell culture, versus SBN. Data represent fold changes of target genes normalized with reference genes (rRNA 18 S and Gapdh), expressed as a percentage of cells seeded on SBN, which were set to 100%. Values represent mean ± SEM. One-way ANOVA and DMS post hoc analyses were performed: * *p* < 0.05 vs. SBN, # vs. SBP2, and vs. SBP6.

**Table 1 jcm-08-02159-t001:** Peptide sequence of the different NuPep variants.

NuPep	Peptide Sequence
P2	PLV PSQ PLV PSQ PLV PSQ PQ PPLPP
P6	PHQ PMQ PQP PVH PMQ PLP PQ PPLPP

**Table 2 jcm-08-02159-t002:** Primers used in the real-time PCR of reference and target genes. Bone morphogenetic protein 2 (BMP-2), Bone sialoprotein (Bsp), Collagen type 1 (Coll-I), Fibronectin 1 (Fn1), Integrin beta 3 (Itgb3), Osteocalcin (Oc), Osterix (Osx) and glyceraldehyde-3-phosphate dehydrogenase (Gapdh).

Gene	Sense Primer Sequence	Antisense Primer Sequence
BMP-2	5′- GCTCCACAAACGAGAAAAGC -3′	5′- AGCAAGGGGAAAAGGACACT -3′
Bonesialoprotein	5’- GAAAATGGAGACGGCGATAG -3´	5´- ACCCGAGAGTGTGGAAAGTG -3´
Collagen-I	5´- AGAGCATGACCGATGGATTC -3´	5´- CCTTCTTGAGGTTGCCAGTC -3´
Fibronectin	5′- GCTGCCAGGAGACAGCCGTG -3′	5′- GTCTTGCCGCCCTTCGGTGG -3′
Integrin beta1	5′- AGCAGGCGTGGTTGCTGGAA -3′	5′- TTTCACCCGTGTCCCACTTGGC -3′
Osteocalcin	5´- CCGGGAGCAGTGTGAGCTTA -3´	5´- TAGATGCGTTTGTAGGCGGTC -3´
Osterix	5´- ACTGGCTAGGTGGTGGTCAG -3´	5´- GGTAGGGAGCTGGGTTAAGG -3´
18S rRNA	5′- GTAACCCGTTGAACCCCATT -3′	5´- CCATCCAATCGGTAGTAGCG -3´
GAPDH	5´- ACCCAGAAGACTGTGGATGG -3´	5´- CACATTGGGGGTAGGAACAC -3´

**Table 3 jcm-08-02159-t003:** Mechanical performances of SBP were not significantly difference to the reported value for SBN [33]. Values presented as mean values with standard deviation (±SD, *n =* 6).

**Torsion**	**Max. Torque (Nmm)**	**Max. Stress (MPa)**	**Max. Strain (%)**	**Torsion Elastic Modulus (MPa)**
	**15050.4 ± 294.9**	**43610.0 ± 4.40**	**43682.0 ± 0.90**	**490.6 ± 103.70**
**Bending**	**Max. Force (N)**	**Max. Stress (MPa)**	**Max. Strain (%)**	**Flexural Modulus (MPa)**
	100.3 ± 17.40	23.8 ± 4.2	7.6 ± 0.90	340.6 ± 63.1
**Compression**	**Max. Force (N)**	**Max. Stress (MPa)**	**Max. Strain (%)**	**Elastic Modulus (MPa)**
	1914.2 ± 590.60	25.8 ± 7.80	2.2 ± 0.40	1245.7 ± 225.90

**Table 4 jcm-08-02159-t004:** Micro-CT results of SmartBone^®^ (SBN) and SBP. Values presented as mean values with standard deviation (±SD, *n =* 3). No significant differences between the groups was observed.

	Open Porosity (%)	Pore Size (µm)	Structural Thickness (µm)	Surface to Volume Ratio (1/µm)
SBN	68.9 ± 2.8	378.3 ± 145.0	148.2 ± 45.6	30.7 ± 1.5
SBP2	70.3 ± 2.9	385.9 ± 147.9	166.0 ± 51.1	27.9 ± 1.3
SBP6	71.7 ± 3.0	393.6 ± 150.9	163.0 ± 50.2	28.8 ± 1.4
SBP2-P6	70.8 ± 2.9	401.5 ± 153.9	157.1 ± 48.3	29.9 ± 1.4
SBP6–1/10	68.6 ± 2.8	454.0 ± 174.0	152.6 ± 47.0	31.6 ± 1.5

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
