# Peer review of "Biomimetic Biomolecules in Next Generation Xeno-Hybrid Bone Graft Material Show Enhanced In Vitro Bone Cells Response"

_jcm, 2019, doi:10.3390/jcm8122159_

Round 1
Reviewer 1 Report
In this manuscript, the authors proposed a strategy to improve the biomineralization capacity of an already existing product for bone repair applications, the SmartBone. To this end, they have successfully loaded the scaffolds with proline-based IDPs (P2 and P6) and studied the effects of such modifications in vitro, using a murine osteoblastic cell line. This study seems to represent the first line of screening in the context of a larger project with multiple aims and deliverables. The manuscript follows a logical sequence, and all the experiments seem to have been performed systematically and with high quality standards. However I believe that further clarification/investigation of some topics is needed. Also, some parts of the main text need to be streamlined to keep the focus of the reader (avoid redundancies).
Major criticism/revisions:
The most promising results in terms of osteogenic activity were achieved in condition SBP6-1/10. Why testing only P6 at 1/10 of target concentration? Was it an educated guess? The rationale for testing this condition must be better explained. Seeding efficiency should also be calculated: (initial #cells/ #cells in scaffold) x100. Regarding the number of cells, the total RNA amount usually is not used to infer about the number of cells; it’s instead a parameter describing their transcriptome kinetics. Therefore, conclusions taken from picture 7 regarding the number of cells seem overreaching. Cell numbers could be extrapolated from the amount of dsDNA. The use of confocal microscopy to visualize cells at 60 µm deep is interesting but does not reflect the situation in the center of the scaffold. The scaffold has to be cut, and the cells visualized, with the same staining, in deeper regions. Moreover, this type of analysis is not quantitative, and inferring about the number of cells from this data set should be done with care. If possible, please perform a semi-quantification of the number of cells? Is purified water a good system to evaluate the kinetics of NuPep release? Is there a reference in the literature that could be added? On page 15, line 390, authors claimed homogeneous distribution of the peptide. Is this claim supported by data? One major claim is that NuPep could potentially enhance the biomineralization capacity of the scaffolds. I have not seen any data suggesting that biomineralization was stimulated in SBP as compared to the SBN, only a significant increase in osteocalcin gene expression in condition P6-1/10. I believe the authors should, at least, perform immunohistochemistry to confirm upregulation of this protein. Moreover, quantification of the mineralization capacity in vitro (calcium phosphate) could be used to assess the biomineralization capacity of these scaffolds.
Minor revision:
Please, consider adding the concept of SmartBonePep in the abstract. Elaborate on the advantage of using biomolecules tolerant to organic solvent in the introduction Are the NuPep mostly disordered or fully disordered proteins? Please, elaborate on the use of a murine osteoblastic cell line and not a human cell line, such as hFOB.Author Response
Reviewer #1
In this manuscript, the authors proposed a strategy to improve the biomineralization capacity of an already existing product for bone repair applications, the SmartBone. To this end, they have successfully loaded the scaffolds with proline-based IDPs (P2 and P6) and studied the effects of such modifications in vitro, using a murine osteoblastic cell line. This study seems to represent the first line of screening in the context of a larger project with multiple aims and deliverables. The manuscript follows a logical sequence, and all the experiments seem to have been performed systematically and with high quality standards. However I believe that further clarification/investigation of some topics is needed. Also, some parts of the main text need to be streamlined to keep the focus of the reader (avoid redundancies).
-Yes, it is correct that this is the first paper in the row of many studies for these bone graft. We have already performed in vitro culture with both human osteoblasts and stem cells from several donors as well as in vivo studies. This will be published in separate papers.
Major criticism/revisions:
The most promising results in terms of osteogenic activity were achieved in condition SBP6-1/10. Why testing only P6 at 1/10 of target concentration? Was it an educated guess? The rationale for testing this condition must be better explained.
-Thank you for raising this issue as this has not been properly explained. The concentrations aimed for these bone graft material is a concentration similar to a commercial product called Emdogain® which is sold by the Institute Straumann AG. This is an injectable hydrogel with enamel matrix protein (EMD) extracted from pigs and is used for dental regenerations. This product have shown in several multi centre studies to be able to regenerated both periodontal ligament, cementum and bone. The molar concentrations of EMD is the same as the molar concentration in the P2 and P6 in these bone graft. We have changed the text according to make this clearer by added this sentence:
“The target concentration is set by previous studies [1-4] with NuPep and is equivalent to molar concentration of EMD in Emdogain®.”
Seeding efficiency should also be calculated: (initial #cells/ #cells in scaffold) x100. Regarding the number of cells, the total RNA amount usually is not used to infer about the number of cells; it’s instead a parameter describing their transcriptome kinetics. Therefore, conclusions taken from picture 7 regarding the number of cells seem overreaching. Cell numbers could be extrapolated from the amount of dsDNA. The use of confocal microscopy to visualize cells at 60 µm deep is interesting but does not reflect the situation in the center of the scaffold. The scaffold has to be cut, and the cells visualized, with the same staining, in deeper regions. Moreover, this type of analysis is not quantitative, and inferring about the number of cells from this data set should be done with care. If possible, please perform a semi-quantification of the number of cells?
For these experiments, we did not perform any quantitative measure of the number of cells attached to the scaffold. However, when setting up the agitating seeding method for the same cell line using scaffolds [5] we determined that after seeding, approximately half of the cells initially seeded attached to the scaffold and the other half attached to the tissue culture plastic of the wells. This agitated seeding method favours a better attachment and superior distribution of cells throughout the 3D scaffold structure.
As for the amount of RNA, we think that its indicative of the amount of cells, since a typical mammalian cell contains 10–30 pg total RNA, being the majority of RNA molecules tRNAs and rRNAs. Accounting mRNA for only 1–5% of the total cellular RNA. However, in agreement with the referee we have softened our conclusions from picture 7.
Unfortunately, we could not perform further fluorescence staining of the scaffolds, thus we have changed the discussion in the manuscript to avoid misinterpretation.
Is purified water a good system to evaluate the kinetics of NuPep release? Is there a reference in the literature that could be added?
-Yes, of course it would be more clinical relevant to do the release kinetics in solution closer to bodily fluids. The release might be different when other peptides and proteins are present. In order to avoid misinterpretation as other proteins might interact with our peptide, it was chosen to use purified water. We have added a sentence regarding the drawback of using such fluids.
“One must, however, mentioned that these releases where done in purify water and that one could expect a different profile once placed in in vivo due to possible interactions with ions and proteins.”
On page 15, line 390, authors claimed homogeneous distribution of the peptide. Is this claim supported by data?
-Yes, it is is. We have added the sentence “ …..as one observed a uniformed spread during EDX analysis (data not shown). We can provide these EDX results, but felt that the manuscript had enough figures already.
One major claim is that NuPep could potentially enhance the biomineralization capacity of the scaffolds. I have not seen any data suggesting that biomineralization was stimulated in SBP as compared to the SBN, only a significant increase in osteocalcin gene expression in condition P6-1/10. I believe the authors should, at least, perform immunohistochemistry to confirm upregulation of this protein. Moreover, quantification of the mineralization capacity in vitro (calcium phosphate) could be used to assess the biomineralization capacity of these scaffolds.
Yes, we do agree that the current methodology is not sufficient to claim any enhances biomineralization. This has been shown previsously [1-4,6], but not in this study. We are currently performing immunohistochemistry to confirm this from an in vivo studies, however this will be in a separate manuscript. The text has been amended accordingly and the word “enhanced biomineralization” has been removed
Minor revision:
Please, consider adding the concept of SmartBonePep in the abstract. Elaborate on the advantage of using biomolecules tolerant to organic solvent in the introduction Are the NuPep mostly disordered or fully disordered proteins?
-Thanks for this feedback. We have tried, with constrains on the number of words in the abstract to rewrite and restructure the abstract, and do hope that the review finds the current version more satisfactory
Please, elaborate on the use of a murine osteoblastic cell line and not a human cell line, such as hFOB.
-We have added a section on the advantages and disadvantages of using murine osteoblastic cell line. It is indeed important to address the shortcoming of only evaluating new materials with cell lines.
Review #2
The aim of the present manuscript was to develop and assess the in vitro performances of a xeno-hybrid composite bone graft, enriched with proline-based peptides, also named SmartBonePep® (SBP) and to verify that the added peptide had a positive outcome on cellular response.
The paper is very well written and presented with accuracy.
-We are very grateful for this comment and do hope that the audience out there is also of the same opinion.
The study is innovative both for the topic and for the methodology. The topic is in line with the journal aim. The study is novel for the topic and it appears well structured, correctly carried out and written without logical or factual errors. Data reported in the Methods section are appropriate and precisely described; they appear to be reproducible. Results are reported clearly and adequately supported by images. The images are clear and indicative of the content.
The Conclusions are correctly stated and supported by the findings obtained from the present study.
However, there are some parts that should be reorganized and improved as suggested below:
- In the Title the Authors should add “An in Vitro study”
Title is amended
-The Abstract should be rewritten in a clearer and more precise form. By reading the Abstract,the study design carried out is not clear. This section should be rewritten in a more schematic form to make easier for the reader to understand.
The abstract has been rewritten
-In the Introduction section the following sentence: “This means that it is necessary to develop a device that regulates the bone regeneration in a manner that stimulates biomineralization at a high rate, but only when necessary.” The Authors should better clarify what they mean by “when necessary”.
-Sentence has been rewritten for clarity
- “Rubert et al. had great success with structural and biological activity for synthetic proline-rich peptides and showed that by tuning the peptide structure they could successfully produce a peptide, which better and more specifically initiated osteoblastic differentiation compared to the EMD” The Authors should add a description of EMD to better understanding the differences between EMD and proline-rich peptides
-Section has been altered accordingly, we agree that the section was not well prepared.
-In the Introduction section the Authors should clarify the choice of the xeno-hybrid composite bone; they should describe the advantages of this material.
-Sentence has been added
-In the same section could be appropriate to describe the mouse osteoblastic cell line.
The present work was aimed to develop and assess the in vitro performances of a xeno-hybrid composite bone graft, enriched with proline-based peptides, also named SmartBonePep® (SBP) and to verify that the added peptide had a positive outcome on cellular response. A murine osteoblastic cell line MC3T3-E1 (DSMZ, Braunschweig, Germany) was used for the in vitro experiments. In vitro studies with large samples numbers can be screened rapidly with the fast growing MC3T3-E1 cell line and this cell line provides reproducible results and is robust and well applicable for screening purposes [7]. The cell line is much used in initial studies of biomaterials intended for use in bone application [8-12], nevertheless, for studies concerning detailed involved bone formation mechanisms, cell lines are not sufficient. In comparison, primary cells have a better indication for the expected effects in vivo, but are also less stable and employ different pathways as well as differences in expression of proteins and mRNA levels [13].
In the Experimental Section:
-The Authors should describe and clarify why the disks used for mechanical tests had different measures than the disks used for cell cultures;
Yes, we agree that this might raise confusion. We have added these two sentences:
The larger samples for cell culture was made to ensure better cell spreading. The cubes for mechanical testing was made as these cubes are similar in size to those BS used in the clinic.
References used in the comments
Riksen, E.A.; Petzold, C.; Brookes, S.; Lyngstadaas, S.P.; Reseland, J.E. Human osteoblastic cells discriminate between 20-kDa amelogenin isoforms. European journal of oral sciences 2011, 119, 357-365, doi:10.1111/j.1600-0722.2011.00912.x. Rubert, M.; Pullisaar, H.; Gómez-Florit, M.; Ramis, J.M.; Tiainen, H.; Haugen, H.J.; Lyngstadaas, S.P.; Monjo, M. Effect of TiO2 scaffolds coated with alginate hydrogel containing a proline-rich peptide on osteoblast growth and differentiation in vitro. J Biomed Mater Res A 2013, 101A, 1768-1777, doi:10.1002/jbm.a.34458. Rubert, M.; Ramis, J.M.; Vondrasek, J.; Gaya, A.; Lyngstadaas, S.P.; Monjo, M. Synthetic Peptides Analogue to Enamel Proteins Promote Osteogenic Differentiation of MC3T3-E1 and Mesenchymal Stem Cells. J Biomater Tiss Eng 2011, 1, 198-209, doi:10.1166/jbt.2011.1018. Villa, O.; Brookes, S.J.; Thiede, B.; Heijl, L.; Lyngstadaas, S.P.; Reseland, J.E. Subfractions of enamel matrix derivative differentially influence cytokine secretion from human oral fibroblasts. Journal of Tissue Engineering 2015, 24, doi:http://dx.doi.org10.1177/2041731415575857. Gomez-Florit, M.; Rubert, M.; Ramis, J.M.; Haugen, H.J.; Tiainen, H.; Lyngstadaas, S.P.; Monjo, M. TiO2 Scaffolds Sustain Differentiation of MC3T3-E1 Cells. J Biomater Tiss Eng 2012, 2, 336-344, doi:10.1166/jbt.2012.1055. Petzold, C.; Monjo, M.; Rubert, M.; Reinholt, F.P.; Gomez-Florit, M.; Ramis, J.M.; Ellingsen, J.E.; Lyngstadaas, S.P. Effect of Proline-Rich Synthetic Peptide-Coated Titanium Implants on Bone Healing in a Rabbit Model. Int J Oral Max Impl 2013, 28, E547-E555. Beck, G.R.; Sullivan, E.C.; Moran, E.; Zerler, B. Relationship between alkaline phosphatase levels, osteopontin expression, and mineralization in differentiating MC3T3‐E1 osteoblasts. J Cell Biochem 1998, 68, 269-280. Shu, R.; McMullen, R.; Baumann, M.; McCabe, L. Hydroxyapatite accelerates differentiation and suppresses growth of MC3T3‐E1 osteoblasts. Journal of Biomedical Materials Research Part A: An Official Journal of The Society for Biomaterials, The Japanese Society for Biomaterials, and The Australian Society for Biomaterials and the Korean Society for Biomaterials 2003, 67, 1196-1204. St-Pierre, J.-P.; Gauthier, M.; Lefebvre, L.-P.; Tabrizian, M. Three-dimensional growth of differentiating MC3T3-E1 pre-osteoblasts on porous titanium scaffolds. Biomaterials 2005, 26, 7319-7328. Eisenbarth, E.; Velten, D.; Müller, M.; Thull, R.; Breme, J. Biocompatibility of β-stabilizing elements of titanium alloys. Biomaterials 2004, 25, 5705-5713. Lamolle, S.F.; Monjo, M.; Rubert, M.; Haugen, H.J.; Lyngstadaas, S.P.; Ellingsen, J.E. The effect of hydrofluoric acid treatment of titanium surface on nanostructural and chemical changes and the growth of MC3T3-E1 cells. Biomaterials 2009, 30, 736-742, doi:10.1016/j.biomaterials.2008.10.052. Walter, M.S.; Frank, M.J.; Satue, M.; Monjo, M.; Ronold, H.J.; Lyngstadaas, S.P.; Haugen, H.J. Bioactive implant surface with electrochemically bound doxycycline promotes bone formation markers in vitro and in vivo. Dent Mater 2014, 30, 200-214, doi:10.1016/j.dental.2013.11.006. Pan, C.; Kumar, C.; Bohl, S.; Klingmueller, U.; Mann, M. Comparative proteomic phenotyping of cell lines and primary cells to assess preservation of cell type-specific functions. Mol Cell Proteomics 2009, 8, 443-450.
Reviewer 2 Report
The aim of the present manuscript was to develop and assess the in vitro performances of a xeno-hybrid composite bone graft, enriched with proline-based peptides, also named SmartBonePep® (SBP) and to verify that the added peptide had a positive outcome on cellular response.
The paper is very well written and presented with accuracy.
The study is innovative both for the topic and for the methodology. The topic is in line with the journal aim. The study is novel for the topic and it appears well structured, correctly carried out and written without logical or factual errors. Data reported in the Methods section are appropriate and precisely described; they appear to be reproducible. Results are reported clearly and adequately supported by images. The images are clear and indicative of the content.
The Conclusions are correctly stated and supported by the findings obtained from the present study.
However, there are some parts that should be reorganized and improved as suggested below:
- In the Title the Authors should add “An in Vitro study”
-The Abstract should be rewritten in a clearer and more precise form. By reading the Abstract,
the study design carried out is not clear. This section should be rewritten in a more schematic form to make easier for the reader to understand.
-In the Introduction section the following sentence: “This means that it is necessary to develop a device that regulates the bone regeneration in a manner that stimulates biomineralization at a high rate, but only when necessary.” The Authors should better clarify what they mean by “when necessary”.
- “Rubert et al. had great success with structural and biological activity for synthetic proline-rich peptides and showed that by tuning the peptide structure they could successfully produce a peptide, which better and more specifically initiated osteoblastic differentiation compared to the EMD” The Authors should add a description of EMD to better understanding the differences between EMD and proline-rich peptides
-In the Introduction section the Authors should clarify the choice of the xeno-hybrid composite bone; they should describe the advantages of this material.
-In the same section could be appropriate to describe the mouse osteoblastic cell line.
In the Experimental Section:
-The Authors should describe and clarify why the disks used for mechanical tests had different measures than the disks used for cell cultures;
Author Response
Reviewer #1
In this manuscript, the authors proposed a strategy to improve the biomineralization capacity of an already existing product for bone repair applications, the SmartBone. To this end, they have successfully loaded the scaffolds with proline-based IDPs (P2 and P6) and studied the effects of such modifications in vitro, using a murine osteoblastic cell line. This study seems to represent the first line of screening in the context of a larger project with multiple aims and deliverables. The manuscript follows a logical sequence, and all the experiments seem to have been performed systematically and with high quality standards. However I believe that further clarification/investigation of some topics is needed. Also, some parts of the main text need to be streamlined to keep the focus of the reader (avoid redundancies).
-Yes, it is correct that this is the first paper in the row of many studies for these bone graft. We have already performed in vitro culture with both human osteoblasts and stem cells from several donors as well as in vivo studies. This will be published in separate papers.
Major criticism/revisions:
The most promising results in terms of osteogenic activity were achieved in condition SBP6-1/10. Why testing only P6 at 1/10 of target concentration? Was it an educated guess? The rationale for testing this condition must be better explained.
-Thank you for raising this issue as this has not been properly explained. The concentrations aimed for these bone graft material is a concentration similar to a commercial product called Emdogain® which is sold by the Institute Straumann AG. This is an injectable hydrogel with enamel matrix protein (EMD) extracted from pigs and is used for dental regenerations. This product have shown in several multi centre studies to be able to regenerated both periodontal ligament, cementum and bone. The molar concentrations of EMD is the same as the molar concentration in the P2 and P6 in these bone graft. We have changed the text according to make this clearer by added this sentence:
“The target concentration is set by previous studies [1-4] with NuPep and is equivalent to molar concentration of EMD in Emdogain®.”
Seeding efficiency should also be calculated: (initial #cells/ #cells in scaffold) x100. Regarding the number of cells, the total RNA amount usually is not used to infer about the number of cells; it’s instead a parameter describing their transcriptome kinetics. Therefore, conclusions taken from picture 7 regarding the number of cells seem overreaching. Cell numbers could be extrapolated from the amount of dsDNA. The use of confocal microscopy to visualize cells at 60 µm deep is interesting but does not reflect the situation in the center of the scaffold. The scaffold has to be cut, and the cells visualized, with the same staining, in deeper regions. Moreover, this type of analysis is not quantitative, and inferring about the number of cells from this data set should be done with care. If possible, please perform a semi-quantification of the number of cells?
For these experiments, we did not perform any quantitative measure of the number of cells attached to the scaffold. However, when setting up the agitating seeding method for the same cell line using scaffolds [5] we determined that after seeding, approximately half of the cells initially seeded attached to the scaffold and the other half attached to the tissue culture plastic of the wells. This agitated seeding method favours a better attachment and superior distribution of cells throughout the 3D scaffold structure.
As for the amount of RNA, we think that its indicative of the amount of cells, since a typical mammalian cell contains 10–30 pg total RNA, being the majority of RNA molecules tRNAs and rRNAs. Accounting mRNA for only 1–5% of the total cellular RNA. However, in agreement with the referee we have softened our conclusions from picture 7.
Unfortunately, we could not perform further fluorescence staining of the scaffolds, thus we have changed the discussion in the manuscript to avoid misinterpretation.
Is purified water a good system to evaluate the kinetics of NuPep release? Is there a reference in the literature that could be added?
-Yes, of course it would be more clinical relevant to do the release kinetics in solution closer to bodily fluids. The release might be different when other peptides and proteins are present. In order to avoid misinterpretation as other proteins might interact with our peptide, it was chosen to use purified water. We have added a sentence regarding the drawback of using such fluids.
“One must, however, mentioned that these releases where done in purify water and that one could expect a different profile once placed in in vivo due to possible interactions with ions and proteins.”
On page 15, line 390, authors claimed homogeneous distribution of the peptide. Is this claim supported by data?
-Yes, it is is. We have added the sentence “ …..as one observed a uniformed spread during EDX analysis (data not shown). We can provide these EDX results, but felt that the manuscript had enough figures already.
One major claim is that NuPep could potentially enhance the biomineralization capacity of the scaffolds. I have not seen any data suggesting that biomineralization was stimulated in SBP as compared to the SBN, only a significant increase in osteocalcin gene expression in condition P6-1/10. I believe the authors should, at least, perform immunohistochemistry to confirm upregulation of this protein. Moreover, quantification of the mineralization capacity in vitro (calcium phosphate) could be used to assess the biomineralization capacity of these scaffolds.
Yes, we do agree that the current methodology is not sufficient to claim any enhances biomineralization. This has been shown previsously [1-4,6], but not in this study. We are currently performing immunohistochemistry to confirm this from an in vivo studies, however this will be in a separate manuscript. The text has been amended accordingly and the word “enhanced biomineralization” has been removed
Minor revision:
Please, consider adding the concept of SmartBonePep in the abstract. Elaborate on the advantage of using biomolecules tolerant to organic solvent in the introduction Are the NuPep mostly disordered or fully disordered proteins?
-Thanks for this feedback. We have tried, with constrains on the number of words in the abstract to rewrite and restructure the abstract, and do hope that the review finds the current version more satisfactory
Please, elaborate on the use of a murine osteoblastic cell line and not a human cell line, such as hFOB.
-We have added a section on the advantages and disadvantages of using murine osteoblastic cell line. It is indeed important to address the shortcoming of only evaluating new materials with cell lines.
Review #2
The aim of the present manuscript was to develop and assess the in vitro performances of a xeno-hybrid composite bone graft, enriched with proline-based peptides, also named SmartBonePep® (SBP) and to verify that the added peptide had a positive outcome on cellular response.
The paper is very well written and presented with accuracy.
-We are very grateful for this comment and do hope that the audience out there is also of the same opinion.
The study is innovative both for the topic and for the methodology. The topic is in line with the journal aim. The study is novel for the topic and it appears well structured, correctly carried out and written without logical or factual errors. Data reported in the Methods section are appropriate and precisely described; they appear to be reproducible. Results are reported clearly and adequately supported by images. The images are clear and indicative of the content.
The Conclusions are correctly stated and supported by the findings obtained from the present study.
However, there are some parts that should be reorganized and improved as suggested below:
- In the Title the Authors should add “An in Vitro study”
Title is amended
-The Abstract should be rewritten in a clearer and more precise form. By reading the Abstract,the study design carried out is not clear. This section should be rewritten in a more schematic form to make easier for the reader to understand.
The abstract has been rewritten
-In the Introduction section the following sentence: “This means that it is necessary to develop a device that regulates the bone regeneration in a manner that stimulates biomineralization at a high rate, but only when necessary.” The Authors should better clarify what they mean by “when necessary”.
-Sentence has been rewritten for clarity
- “Rubert et al. had great success with structural and biological activity for synthetic proline-rich peptides and showed that by tuning the peptide structure they could successfully produce a peptide, which better and more specifically initiated osteoblastic differentiation compared to the EMD” The Authors should add a description of EMD to better understanding the differences between EMD and proline-rich peptides
-Section has been altered accordingly, we agree that the section was not well prepared.
-In the Introduction section the Authors should clarify the choice of the xeno-hybrid composite bone; they should describe the advantages of this material.
-Sentence has been added
-In the same section could be appropriate to describe the mouse osteoblastic cell line.
The present work was aimed to develop and assess the in vitro performances of a xeno-hybrid composite bone graft, enriched with proline-based peptides, also named SmartBonePep® (SBP) and to verify that the added peptide had a positive outcome on cellular response. A murine osteoblastic cell line MC3T3-E1 (DSMZ, Braunschweig, Germany) was used for the in vitro experiments. In vitro studies with large samples numbers can be screened rapidly with the fast growing MC3T3-E1 cell line and this cell line provides reproducible results and is robust and well applicable for screening purposes [7]. The cell line is much used in initial studies of biomaterials intended for use in bone application [8-12], nevertheless, for studies concerning detailed involved bone formation mechanisms, cell lines are not sufficient. In comparison, primary cells have a better indication for the expected effects in vivo, but are also less stable and employ different pathways as well as differences in expression of proteins and mRNA levels [13].
In the Experimental Section:
-The Authors should describe and clarify why the disks used for mechanical tests had different measures than the disks used for cell cultures;
Yes, we agree that this might raise confusion. We have added these two sentences:
The larger samples for cell culture was made to ensure better cell spreading. The cubes for mechanical testing was made as these cubes are similar in size to those BS used in the clinic.
References used in the comments
Riksen, E.A.; Petzold, C.; Brookes, S.; Lyngstadaas, S.P.; Reseland, J.E. Human osteoblastic cells discriminate between 20-kDa amelogenin isoforms. European journal of oral sciences 2011, 119, 357-365, doi:10.1111/j.1600-0722.2011.00912.x. Rubert, M.; Pullisaar, H.; Gómez-Florit, M.; Ramis, J.M.; Tiainen, H.; Haugen, H.J.; Lyngstadaas, S.P.; Monjo, M. Effect of TiO2 scaffolds coated with alginate hydrogel containing a proline-rich peptide on osteoblast growth and differentiation in vitro. J Biomed Mater Res A 2013, 101A, 1768-1777, doi:10.1002/jbm.a.34458. Rubert, M.; Ramis, J.M.; Vondrasek, J.; Gaya, A.; Lyngstadaas, S.P.; Monjo, M. Synthetic Peptides Analogue to Enamel Proteins Promote Osteogenic Differentiation of MC3T3-E1 and Mesenchymal Stem Cells. J Biomater Tiss Eng 2011, 1, 198-209, doi:10.1166/jbt.2011.1018. Villa, O.; Brookes, S.J.; Thiede, B.; Heijl, L.; Lyngstadaas, S.P.; Reseland, J.E. Subfractions of enamel matrix derivative differentially influence cytokine secretion from human oral fibroblasts. Journal of Tissue Engineering 2015, 24, doi:http://dx.doi.org10.1177/2041731415575857. Gomez-Florit, M.; Rubert, M.; Ramis, J.M.; Haugen, H.J.; Tiainen, H.; Lyngstadaas, S.P.; Monjo, M. TiO2 Scaffolds Sustain Differentiation of MC3T3-E1 Cells. J Biomater Tiss Eng 2012, 2, 336-344, doi:10.1166/jbt.2012.1055. Petzold, C.; Monjo, M.; Rubert, M.; Reinholt, F.P.; Gomez-Florit, M.; Ramis, J.M.; Ellingsen, J.E.; Lyngstadaas, S.P. Effect of Proline-Rich Synthetic Peptide-Coated Titanium Implants on Bone Healing in a Rabbit Model. Int J Oral Max Impl 2013, 28, E547-E555. Beck, G.R.; Sullivan, E.C.; Moran, E.; Zerler, B. Relationship between alkaline phosphatase levels, osteopontin expression, and mineralization in differentiating MC3T3‐E1 osteoblasts. J Cell Biochem 1998, 68, 269-280. Shu, R.; McMullen, R.; Baumann, M.; McCabe, L. Hydroxyapatite accelerates differentiation and suppresses growth of MC3T3‐E1 osteoblasts. Journal of Biomedical Materials Research Part A: An Official Journal of The Society for Biomaterials, The Japanese Society for Biomaterials, and The Australian Society for Biomaterials and the Korean Society for Biomaterials 2003, 67, 1196-1204. St-Pierre, J.-P.; Gauthier, M.; Lefebvre, L.-P.; Tabrizian, M. Three-dimensional growth of differentiating MC3T3-E1 pre-osteoblasts on porous titanium scaffolds. Biomaterials 2005, 26, 7319-7328. Eisenbarth, E.; Velten, D.; Müller, M.; Thull, R.; Breme, J. Biocompatibility of β-stabilizing elements of titanium alloys. Biomaterials 2004, 25, 5705-5713. Lamolle, S.F.; Monjo, M.; Rubert, M.; Haugen, H.J.; Lyngstadaas, S.P.; Ellingsen, J.E. The effect of hydrofluoric acid treatment of titanium surface on nanostructural and chemical changes and the growth of MC3T3-E1 cells. Biomaterials 2009, 30, 736-742, doi:10.1016/j.biomaterials.2008.10.052. Walter, M.S.; Frank, M.J.; Satue, M.; Monjo, M.; Ronold, H.J.; Lyngstadaas, S.P.; Haugen, H.J. Bioactive implant surface with electrochemically bound doxycycline promotes bone formation markers in vitro and in vivo. Dent Mater 2014, 30, 200-214, doi:10.1016/j.dental.2013.11.006. Pan, C.; Kumar, C.; Bohl, S.; Klingmueller, U.; Mann, M. Comparative proteomic phenotyping of cell lines and primary cells to assess preservation of cell type-specific functions. Mol Cell Proteomics 2009, 8, 443-450.
Round 2
Reviewer 1 Report
Dear authors,
Thank you for addressing all my questions and adapting the manuscript according to my suggestions. I believe this new version of the manuscript is more suitable for publication. However, I am still not entirely sure that the total RNA content could be used to suggest an increased number of cells in condition SBP6. Indeed, in the rebuttal, the authors claimed that the amount of RNA per mammalian is variable (3x difference). Considering that an increase in the number of cells is also not apparent in figure 6 and figure 5 – also there is no quantification - I would suggest that conclusions about the number of cells supported by total RNA content are not accurate and should be removed, as well as the figure itself. However, I leave the final decision on this topic to the editor of the JCM.
Kind regards.
Author Response
dearest reviewer,
we have extended our discussion on this matter.
best
